ecology/behaviour/acoustics

Antarctic minke whales, passive acoustic monitoring, Weddell Sea area, Greenwich area, Elephant Island, bio-duck call

**Author for correspondence:**
Diego Filun
e-mail: diego.filun@awi.de

# Frozen verses: Antarctic minke whales (*Balaenoptera bonaerensis*) call predominantly during austral winter

Diego Filun[1,2], Karolin Thomisch[1], Olaf Boebel[1], Thomas Brey[1,2,3], Ana Širović[4], Stefanie Spiesecke[1] and Ilse Van Opzeeland[1,3]

[1]Ocean Acoustics Lab, Alfred-Wegener-Institute Helmholtz-Zentrum für Polar- und Meeresforschung, 27570 Bremerhaven, Germany
[2]Faculty of Biology/Chemistry, University of Bremen, 28359 Bremen, Germany
[3]Helmholtz Institute for Functional Marine Biodiversity (HIFMB), Carl von Ossietzky University, 26129 Oldenburg, Germany
[4]Department of Marine Biology, Texas A&M University at Galveston, Galveston, TX, USA

DF, 0000-0003-2789-596X

The recent identification of the bio-duck call as Antarctic minke whale (AMW) vocalization allows the use of passive acoustic monitoring to retrospectively investigate year-round spatial-temporal patterns in minke whale occurrence in ice-covered areas. Here, we present an analysis of AMW occurrence patterns based on a 9-year passive acoustic dataset (2008–2016) from 21 locations throughout the Atlantic sector of the Southern Ocean (Weddell Sea). AMWs were detected acoustically at all mooring locations from May to December, with the highest presence between August and November (bio-duck calls present at more than 80% of days). At the southernmost recording locations, the bio-duck call was present up to 10 months of the year. Substantial inter-annual variation in the seasonality of vocal activity correlated to variation in local ice concentration. Our analysis indicates that part of the AMW population stays in the Weddell Sea during austral winter. The period with the highest acoustic presence in the Weddell Sea (September–October) coincides with the timing of the breeding season of AMW in lower latitudes. The bio-duck call could therefore play a role in mating, although other behavioural functions of the call cannot be excluded to date.

# 1. Introduction

The Antarctic sea-ice environment constitutes a productive and dynamic habitat for many unique migratory and resident species, ranging from zooplankton, fish and birds to marine mammals [1]. During austral summer when the retreating sea-ice exposes large quantities of patchily distributed krill, high densities of various krill consumers aggregate in the ice edge region to feed [2–7]. However, there is increasing evidence that many species remain in Antarctic waters during winter in spite of heavy sea-ice cover, exploiting food resources under the ice and in open water patches [8–13]. Our knowledge on the ecology of these species in their sea-ice habitat is hampered severely by the logistic restrictions imposed on scientific work in such environments. For Antarctic krill (*Euphausia superba*), one of the world's most abundant species and foundation of the Southern Ocean food web, information on winter distribution and sea-ice habitat preferences is virtually lacking because survey ships are typically insufficiently ice-strengthened [7,14–17]. The distribution of Antarctic minke whales (*Balaenoptera bonaearensis*) (AMW) is another striking knowledge gap. AMWs have a circum-Antarctic distribution and are known to occur within the marginal ice zone (MIZ) and the interior sea-ice pack [18–20]. Most detailed information on the species stems from ship-based observations, i.e. data are limited to the austral summer period and regions that are (seasonally) ice-free. Visual sightings from the region indicate that the species' behaviour and appearance often result in low sightability, making visual data collection on this species challenging [7,21]. As a consequence, AMW abundance estimates based on visual survey data are in many cases spatially and temporally biased [7,22]. Only very sparse information exists on the winter distribution of AMW in the Southern Ocean and how the species is associated with different sea-ice concentrations throughout the year [23–25].

The recent identification of the bio-duck calls as vocalizations produced by AMWs [26] allows to use passive acoustic data to study their occurrence patterns and behaviour of this species. Passive acoustic monitoring offers a versatile technology with which long-term archival data can be collected on sound-producing species using autonomous recording units [27]. This makes passive acoustic recording techniques a suitable tool to remotely monitor the acoustic presence and study marine mammal behaviour. This particularly accounts for polar species given that large parts of their habitats are (seasonally) inaccessible for research vessels (e.g. [4,12,13,28]).

The identification of AMW sounds marked the end of a long-standing riddle as the bio-duck calls had been recorded at numerous locations across the Southern Ocean as well as off the Australian and Namibian coasts [26,29–31]. With the confirmation that the sound stems from AMWs, it is now possible to retrospectively process passive acoustic time series to explore this species' year-round distribution and relation to sea-ice conditions to improve our understanding of this species also across years detecting trends in behaviour and distribution. Improving our knowledge of AMW ecology and behaviour is central to our understanding of the effects of ongoing climatic change on the Antarctic pelagic ecosystem. During the last decades, (sub)surface warming has been shown to drastically affect the marine environment around the Antarctic Peninsula in various ways [32]. Elevated temperatures trigger reductions in the seasonal period and extension of sea-ice-covered areas, leading to a chain of reactions by which also zooplankton assemblages are impacted with anticipated negative feedbacks to the ecology of higher trophic level consumers [33,34]. AMWs are krill consumers, preferring areas with substantial ice cover [7,20,35]. The forecasted changes in sea-ice conditions therefore have the potential to drastically affect AMWs [20]. To understand and predict how the cascading effects of climate change impact higher trophic level species, such as AMWs, baseline information on species' distribution and habitat preferences is crucial to identify and interpret potential climate-mediated shifts in range boundaries. Information on AMW spatial-temporal distribution patterns is furthermore of direct relevance for Southern Ocean ecosystem management in the context of the conservation and management mandate of the International Whaling Commission (IWC) and the Commission for the Conservation of Antarctic Marine Living Resources (CCAMLR). Recently, AMWs were classified as Near Threatened under the Internal Union for Conservation of Nature (IUCN) Red List and under Appendix I of CITES [36]. Given the current uncertainty regarding AMW abundance estimates (IWC 2013) [37], better insights into spatial-temporal movements of this population or its subpopulations may in part be key to improving methods for abundance estimations.

Here, we use multi-year passive acoustic data from a recording network in the Weddell Sea and along the Greenwich meridian to investigate spatial-temporal patterns in the daily acoustic presence of AMWs.

**Table 1.** Locations and recordings parameters of acoustic recorders deployed within the Hybrid Antarctic Float Observation System (HAFOS) array in the Weddell Sea. Recorders sorted by deployment period.

| recording site ID | recorder ID | latitude | longitude | deployment depth (m) | sampling frequency (kHz) | sampling scheme (min/min) |
|---|---|---|---|---|---|---|
| GW1 | AURAL | 68 59.74 S | 000 00.17 E | 189 | 32.77 | 5/240 |
| GW2 | AURAL | 66 01.13 S | 000 04.77 E | 206 | 32.77 | 5/240 |
| GW3 | SonoVaults | 59 03.02 S | 000 06.63 E | 1007 | 5.33 | continuous |
| GW4 | SonoVaults | 63 59.56 S | 000 02.65 W | 969 | 5.33 | continuous |
| GW1 | SonoVaults | 66 01.90 S | 000 03.25 E | 934 | 5.33 | continuous |
| EI1 | AURAL | 61 00.88 S | 055 58.53 W | 204 | 32.77 | 5/60 |
| WA1 | SonoVaults | 63 28.84 S | 052 05.77 W | 945 | 5.33 | continuous |
| GW1 | SonoVaults | 68 59.86 S | 000 06.51 W | 934 | 5.33 | continuous |
| WA2 | SonoVaults | 69 03.48 S | 017 23.32 W | 678 | 5.33 | continuous |
| WA7 | SonoVaults | 65 58.09 S | 012 15.12 W | 734 | 5.33 | continuous |
| WA6 | AU0086 | 66 36.70 S | 027 07.31 W | 207 | 32.77 | 5/240 |
| WA5 | SonoVaults | 64 22.94 S | 045 52.12 W | 678 | 5.33 | continuous |
| GW5 | SonoVaults | 59 02.63 S | 000 04.92 E | 1020 | 5.33 | continuous |
| WA4 | SonoVaults | 66 36.45 S | 027 07.26 W | 423 | 5.33 | continuous |
| WA4 | SonoVaults | 66 36.45 S | 027 07.26 W | 678 | 5.33 | continuous |
| WA4 | SonoVaults | 65 37.23 S | 036 25.32 W | 956 | 5.33 | continuous |
| WA3 | SonoVaults | 63 42.09 S | 050 49.61 W | 487 | 5.33 | continuous |
| WA3 | SonoVaults | 63 42.09 S | 050 49.61 W | 723 | 5.33 | continuous |
| GW4 | SonoVaults | 64 00.32 S | 000 00.22 W | 812 | 5.33 | continuous |
| GW2 | SonoVaults | 66 30.71 S | 000 01.51 W | 943 | 5.33 | continuous |
| GW1 | SonoVaults | 68 58.89 S | 000 05.00 W | 869 | 5.33 | continuous |
| WA9 | SonoVaults | 66 30.71 S | 000 06.51 W | 1083 | 5.33 | continuous |
| WA8 | SonoVaults | 70 53.55 S | 028 53.47 W | 330 | 5.33 | continuous |

# 2. Material and methods

## 2.1. Data collection

Passive acoustic data were collected between 2008 and 2016 (9 years) from 21 mooring positions (with varying recording periods, see table 1 for details) throughout the Weddell Sea (WA) and along the Greenwich meridian (GW), covering the area between 59 and 69° S and from 0 to 56° W [13].

Passive autonomous acoustic recorders were attached to oceanographic deep sea moorings of the Hybrid Antarctic Float Observation System (HAFOS) [38]. For this study, two types of autonomous acoustic recorders were used: Autonomous Underwater Recorder for Acoustic Listening, model 2 (AURALs; Multi-Electronique, Quebec) and SonoVaults (Develogic GmbH, Hamburg). All four AURALs recorded with a sampling rate of 32 768 Hz. Three of the AURAL recorders were programmed to collect data 5 min every 4 h (2% duty cycle) and one AURAL was set to record 5 min every hour (8% duty cycle). SonoVaults were programmed to record continuously with a sampling rate of 5333 Hz for data collected during years 2010–2014 and with a sampling rate of 6857 Hz during years 2014–2016 (table 1). Although additional analyses showed that the duty cycles applied underestimated daily AMW acoustic presence during the shoulder seasons, overall presence patterns did not show anomalies compared to nearby recorders collecting data continuously (see electronic supplementary material, appendix S1).

The positions located in the Weddell Sea were sub-divided in four different sectors: Elephant Island (EI), Weddell Sea North (WSN), Weddell Sea South (WSS) and Greenwich Meridian (GW) (figure 1). This sub-division was based on the differences in ice growth patterns across the region [39].

R. Soc. Open Sci. 7: 192112

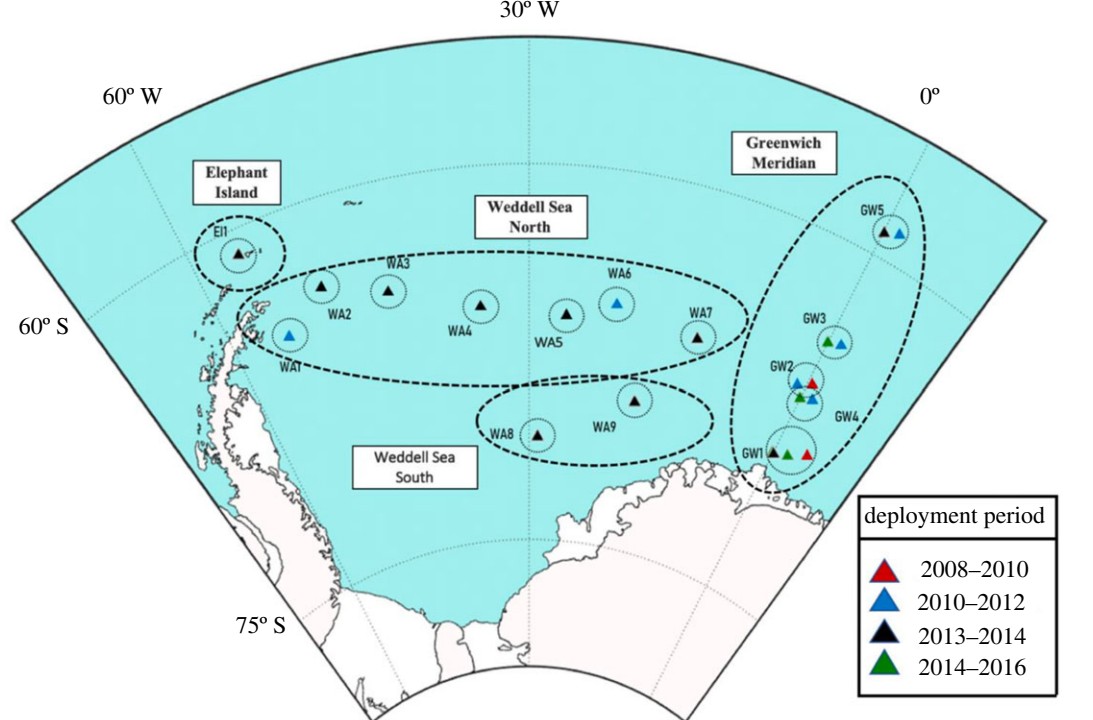

**Figure 1.** Map of the study area in the Weddell Sea. The triangles denote the positions of the passive acoustic recorders and circles represent their detection range (40 km radius). Triangle colour encodes the different deployment periods. Recording sites names are assigned by geographic location (with 'GW', 'WA' and 'EI' indicating the Greenwich meridian, the Weddell Sea and Elephant Island, respectively) and number for distinct recording sites.

## 2.2. Acoustic analyses

The acoustic recordings were processed using the Matlab-based (Mathworks, Natick, MA) custom software program *Triton* [40] to create long-term spectral averages (LTSAs) plots. LTSAs visually represent a time series of averaged spectra [40]. For all data successive spectra, with 1 Hz frequency resolution, were calculated by averaging 60 s of acoustic data. These LTSAs were used for all acoustic data to manually log AMW daily acoustic presence. For the continuous SonoVault recordings, a window size of 1 day of data was inspected to identify AMW signatures. For the duty-cycled AURAL recorders, a four-day window size was used to identify the daily presence of AMWs. When presumed AMW signatures were observed in the LTSAs, a 20 s spectrogram window (overlap = 90%, FFT = 1050) of that time section was inspected visually and aurally to verify the presence of AMW signatures.

We used the bio-duck call as proxy for AMW acoustic presence (see for a description of bio-duck call: [25,26,29,30,41]). The bio-duck is characterized by its repetitive nature, consisting of regular down-sweeps or pulses in series, with most energy located in the 50–300 Hz band (figure 2), although for signals with higher intensity, harmonics occur up to 1 kHz. In this study, a bio-duck call refers to a cluster of down-swept pulses separated by less than 1 s. Different bio-duck sub-types have been described in the literature, mainly basing on the number of pulses within clusters [25,41,42]. Here, no distinction was made between bio-duck sub-types, and all bio-duck calls were pooled to determine daily AMW acoustic presence. Bio-duck calls never occurred alone (i.e. one cluster of calls or one phrase). The minimum sequence duration that we observed was 20 s. All days with AMW acoustic presence therefore at least had a series of 20 s with bio-duck calls over a 24 h period.

Monthly acoustic presence was calculated for all sites from the number of days with AMW acoustic presence per day divided by the number of days with recording effort. To explore inter-annual variability, two positions with 3 years of deployment (GW1 and GW2) were used.

## 2.3. Sea-ice data

The sea-ice concentration data used for this study were extracted from satellite images with a resolution of 6.25 × 6.25 km from the Advanced Microwave Scanning Radiometer for EOS (AMSR-E) satellite sensor

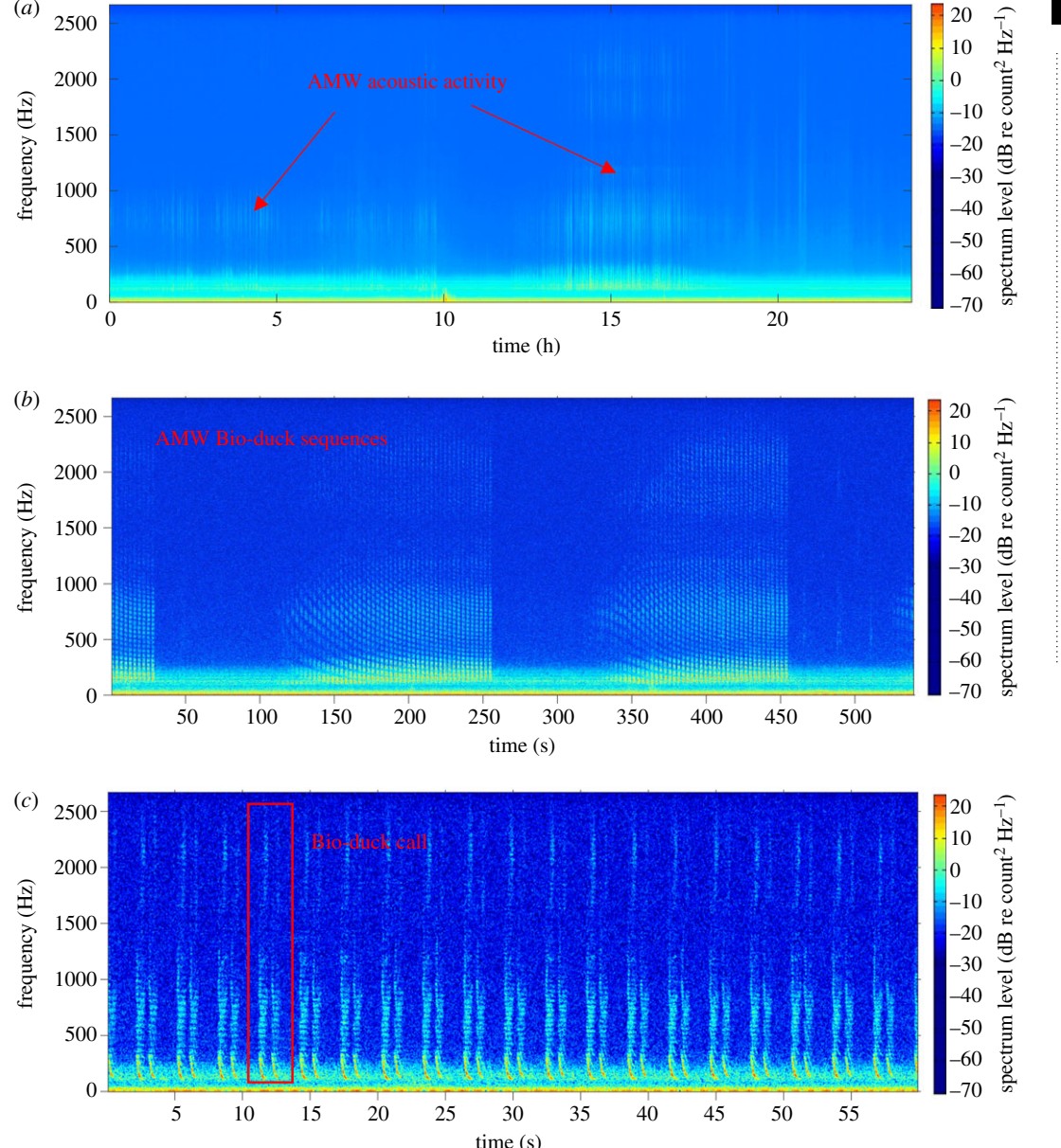

**Figure 2.** (*a*) LTSA of 24 h of acoustic data (sampling rate: 5.333 kHz) with arrows showing AMW acoustic activity (1 min time resolution, 1 Hz frequency resolution). (*b*) Spectrogram zooming into 550 s with the presence of AMW bio-duck sequences (FFT = 1000, overlap 95%). (*c*) Spectrogram zooming into 60 s with various bio-duck calls. Red box indicates a bio-duck call (FFT = 5000, overlap 95%).

[43]. The radius around the mooring over which AMW calls were likely to be detected was determined to extract monthly average sea-ice concentration values only from within this radius. Transmission loss data from empirical analyses on the propagation of signals emitted by an oceanographic (RAFOS) sound source (see [44] for details) were used to calculate the transmission loss for the Greenwich meridian and the Weddell Sea area. The RAFOS signal consisted of an 80 s sweep between 259 Hz and 261 Hz. Transmission loss for this signal was 83 dB ± 4 dB over a distance of 53 km. When transformed into a transmission loss law, it resulted in $16.5 * \log(r)$ to $18.1 * \log10(r)$. Noise levels at this position averaged 74.4 dB ± 4.2 dB. For identification of calls during the manual screening of spectrograms, it was assumed that the SNR of AMW vocalizations would need to exceed 3 dB. Resulting distance estimates for an AMW call with a source level (SL) = 147 dB re 1 µPa [26], ranged between 4 and 27 km. As transmission losses and noise levels were lower during ice-covered periods in austral winter (see [44–46]) and to account for any uncertainties regarding the SLs, or the ability to detect signals with lower SNR in the spectrograms, AMW vocalizations were assumed to be detectable within a radius of 40 km, rather than the calculated 27 km.

To explore how AMW acoustic presence related to sea-ice concentration, ice concentration and acoustic presence data were correlated for every location using a Pearson's correlation. The variability of this relationship with the temporal resolution was explored by correlating AMW presence and sea-ice concentration within moving windows of varying temporal width, i.e. 1, 3, 7, 15, 21, 30 days, to select the best temporal resolution to correlate the acoustic presences with the ice concentration values. Monthly sea-ice concentration data showed highest correlation values ($r = 0.89$)

# 3. Results

Passive acoustic recordings from 21 mooring positions collected between 2008 and 2016 totalling 8176 days were analysed for AMW acoustic presence. AMWs bio-duck signatures were detected on 2777 days (34% of the total number of recording days). AMWs were found to be present at all except the northernmost position (GW5) in this study (table 2).

For two positions (WA3 and WA4), the moorings each contained three recording devices (SonoVaults), which all were programmed to record simultaneously at different depths. Due to the fact that at these positions, all recorders dropped out for some part of the year and we compiled the data from all three recorders for each mooring to obtain one effective year for data analyses on AMW daily presence.

## 3.1. Seasonal and spatial acoustic presence

AMW acoustic presence differed by sectors; at the southernmost positions, AMWs were generally acoustically present longer (i.e. over more months) than at the northern positions (figure 3). At the southern recording positions (i.e. between 70° S and 66° S), acoustic presence (i.e. relative AMW acoustic presence/month) was also higher compared to the northern sites.

AMW bio-duck occurrence exhibited in general a strong seasonal pattern, although AMW acoustic detections varied with location (figure 3). In January and February, AMW acoustic presence was low at all sites, generally increasing from April to July, peaking during August, September and October and decreasing again in November to the sporadic presence in December. At the southernmost position along with GW, AMWs were acoustically present virtually year-round. At the more northern positions along the GW meridian, the onset and increase of AMW acoustic presence was delayed by one month with decreasing recording latitude (figure 3). In July, the percentage of days with AMW presence at all positions along the GW exceeded 60% of days with acoustic recording, peaking in September and October with up to 100% of recording days with the acoustic presence at multiple sites. From the second half of December through March, bio-duck signatures occurred only sporadically in the majority of the recordings from the GW.

In the central Weddell Sea (WSN and WSS), the seasonality of the AMW acoustic presence did not exhibit a clear latitudinal gradient as in the GW sector. At some sites, the onset of acoustic presence was in June whereas at others, AMWs were acoustically present throughout 11 months. For all positions in the Weddell Sea, August, September and October generally exhibited the highest percentage of acoustic presence (greater than 80%) over the recording period (the only exception was position WA1 in August with 42% of days with acoustic presence). The positions WA1, WA3, WA4, WA5, WA6 and WA7 located between 63° S and 66° S (WSN) exhibited more days with acoustic presence compared to the positions located in lower latitudes, WA8 and WA9 (WSS). The positions WA1 and WA3 were the only positions located in the Weddell Sea sector that exhibited AMW acoustic presence during January (figure 3). The positions in WSN had more than 23% of days with acoustic activity in April with the exception of the positions WA3 and WA7 (figure 3). The recorders located in position WA3 stopped collecting data between the months of February and April with only 7 days of effort during May. Position WA7 located closer to the GW sector had the first bio-duck detections during the last days of April.

The position at Elephant Island (EI) had a similar seasonal pattern of AMW, but with fewer days with detections. The first calls were detected during the last days of July (only 2% of days with acoustic detections). August and September were the peak (28% and 29% of days with acoustic activity, respectively). In November, the acoustic presence decreased to just 1% presence.

AMWs were acoustically detected over 9 months (GW4), 11 months (GW2) and 12 months along with the GW sector. The overall percentage of monthly acoustic presence at the northern positions was substantially lower compared to the southern positions (figure 3).

In the sector WSN, one position had an acoustic activity for 11 months (WA1), four positions had between six and seven months of acoustic presence and two positions (WA2 & WA5) had two to four

**Table 2.** Summary table of AMW acoustic presence for all positions. The percentage of days of AMW presence was calculated for every position per year.

| year | position | effort days | days with detections | days excluded | % days AMW presence |
|------|----------|-------------|----------------------|---------------|---------------------|
| 2008 | GW1 | 301 | 164 | 0 | 54 |
|      | GW2 | 296 | 133 | 0 | 44 |
| 2009 | GW1 | 366 | 218 | 0 | 59 |
|      | GW2 | 366 | 151 | 0 | 41 |
| 2010 | GW1 | 353 | 131 | 0 | 37 |
|      | GW2 | 353 | 152 | 0 | 43 |
|      | GW3 | 21 | 0 | 0 | 0 |
| 2011 | GW3 | 213 | 55 | 0 | 25 |
|      | GW4 | 169 | 10 | 0 | 5 |
|      | GW2 | 321 | 155 | 0 | 48 |
|      | WA1 | 268 | 93 | 0 | 34 |
| 2012 | WA2 | 211 | 133 | 0 | 63 |
|      | WA1 | 225 | 21 | 0 | 9 |
| 2013 | WA9 | 233 | 77 | 0 | 33 |
|      | WA3 | 271 | 33 | 0 | 12 |
|      | WA1 | 212 | 136 | 0 | 64 |
|      | WA8 | 288 | 157 | 6 | 54 |
|      | WA4 | 295 | 174 | 0 | 58 |
|      | WA5 | 116 | 32 | 0 | 27 |
|      | WA6 | 292 | 94 | 0 | 32 |
|      | WA7 | 300 | 140 | 21 | 46 |
|      | EI1 | 350 | 26 | 0 | 7 |
|      | WA2 | 221 | 76 | 0 | 34 |
|      | GW1 | 295 | 189 | 0 | 64 |
|      | GW5 | 107 | 77 | 0 | 0 |
| 2014 | EI1 | 365 | 12 | 0 | 33 |
|      | GW4 | 161 | 2 | 0 | 1 |
| 2015 | EI1 | 365 | 12 | 0 | 33 |
|      | GW4 | 161 | 2 | 0 | 1 |
|      | GW1 | 228 | 59 | 0 | 25 |
| 2016 | GW2 | 360 | 140 | 0 | 37 |
|      | EI1 | 93 | 0 | 0 | 0 |

months with acoustic activity. However, recorders moored in these two positions collected data for only eight and five months. In the WSS sector, AMWs were detected eight months in WA8 and four months in the position WA9, where data collection occurred only during seven months. For the mooring positions located in the middle of the Weddell Sea area, there was no apparent north–south gradient in AMW acoustic presence (figure 3).

In the EI position, AMW were detected during only four months and at a low rate in contrast with the other sectors (figure 3).

## 3.2. Inter-annual variability

Two sites with multi-year data, GW1 and GW2, had similar trends in AMW acoustic presence across years (figure 4). It gradually increased from April to austral winter, peaking in spring (August,

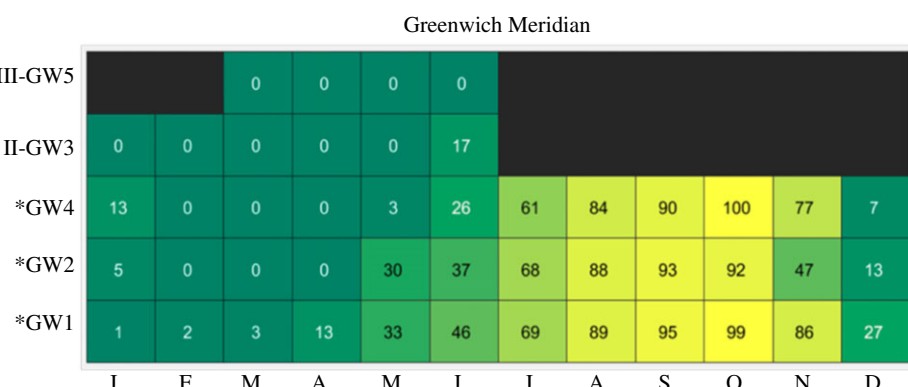

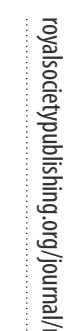

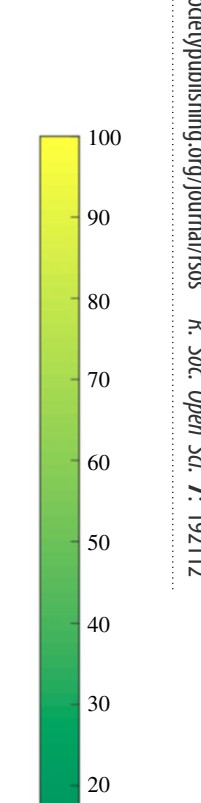

**Figure 3.** Percentage of days with AMW bio-duck presence. Positions sorted by latitude (north to south) in separate sectors: Elephant Island (EI), Weddell Sea North (WSN), Weddell Sea South (WSS) and Greenwich Meridian (GW). For positions for which multi-year data was available (indicated with *), the heatmap displays the average monthly acoustic presence.

September, October and November), exhibiting a consistent pattern between years. Most variability between years occurred at the onset of vocal activity.

## 3.3. Relation to sea-ice concentration

The acoustic presence of AMW was strongly positively correlated with the sea-ice concentration in the Weddell Sea ($r = 0.7$). Days with few detections generally corresponded to days with lower (less than 20%) ice concentration. In general, the acoustic presence of AMW in the Weddell Sea increased with local sea-ice concentration. The positive correlation explains the higher number of days with the acoustic presence with increasing latitude, given that sea-ice is more persistent in the southern recording positions (figure 3). The months with more than 75% of days with acoustic presence typically exhibited local sea-ice concentrations between 75% and 100% in a radius of 40 km around the recording position.

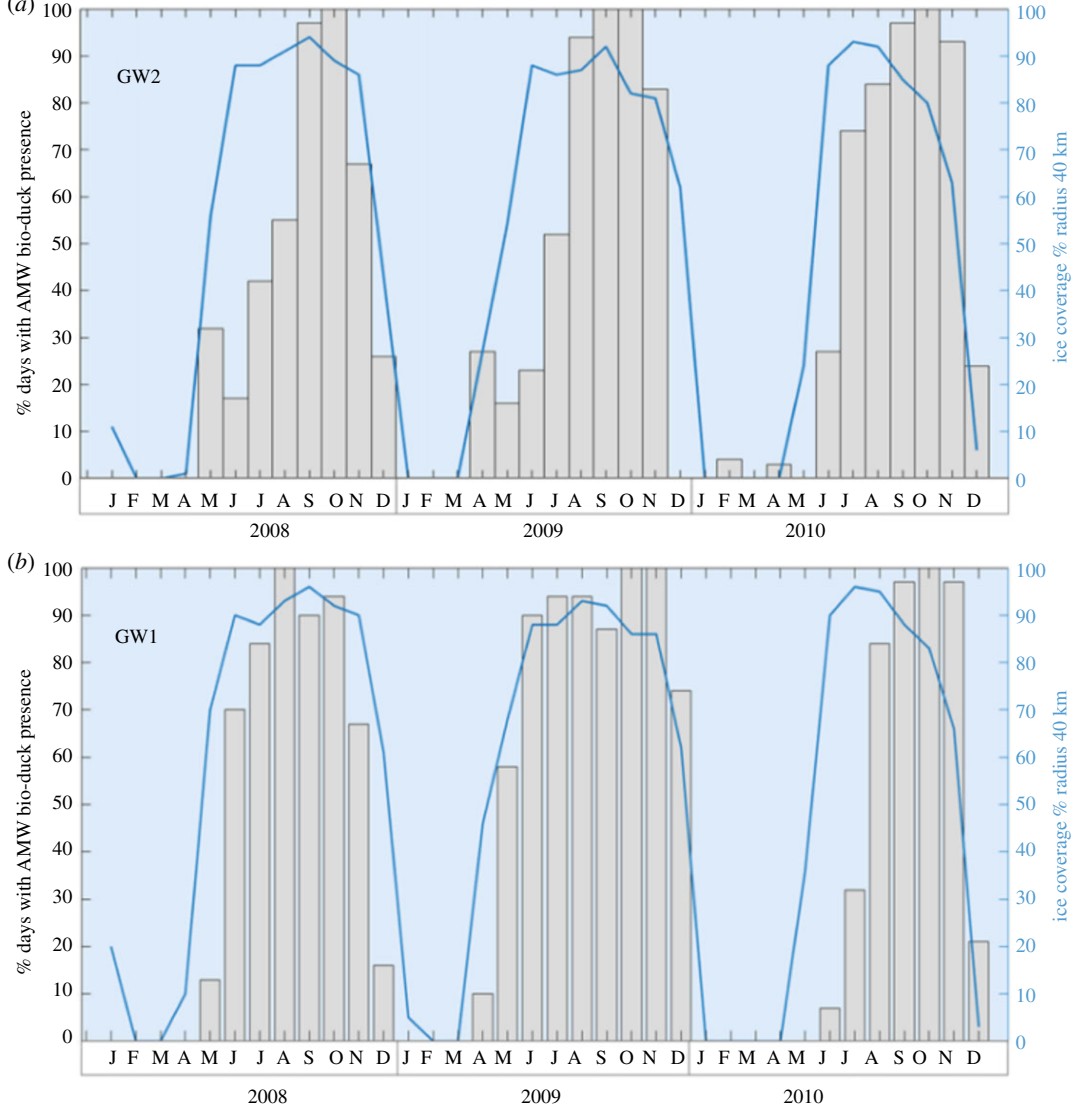

**Figure 4.** Multi-year acoustic presence of AMW in two different positions (GW2 and GW1) correlated with sea-ice coverage in a radius of 40 km. Per cent of days with AMW acoustic presence between 2008 until 2010.

## 4. Discussion

### 4.1. Role of sea-ice

These year-round acoustic observations show that vocally active AMWs exhibit strong affinity with areas featuring dense sea-ice cover. The acoustic observations from winter in the Weddell Sea are consistent with the year-round study west of the Antarctic Peninsula [25], but contrast with what is known about AMW habitat use based on summer sighting data [7,20]. When using icebreaker-supported helicopters for aerial surveys in austral summer, most AMWs were found at the sea-ice edge boundary (defined as the 15% ice concentration contour) during December and January [7]. Herr *et al.* [20] found that during austral summer AMWs occurred close to the sea-ice edge in the Weddell Sea area, and that AMW densities decreased in areas with higher sea-ice concentration values [20]. Contrastingly, the results of our study show none to very little acoustic activity at the recorder locations near the ice edge boundary during December and January. The highest acoustic activity typically occurred in areas and periods of the year with high sea-ice concentrations (greater than 20%). Along the Greenwich meridian, AMW acoustic presence exhibited a latitudinal gradient: at the southernmost positions, AMWs were acoustically present over a longer period (9 to 12 months) compared to the more northern positions. This latitudinal pattern also reflects the strong association between AMW acoustic presence and sea-ice; positions GW1 and GW2 located further south generally are ice covered early (April and May), whereas recording sites III-GW5

and II-GW3 further north along the Greenwich meridian are generally still ice-free during these months. The anomaly between 66° S and 64° S in the latitudinal distribution can be explained due to the presence of a large opening in the sea-ice cover of the Weddell Sea, also known as the Weddell Polynya [47]. AMW acoustic absence in the position GW5 is probably explained by the fact that the recorder only collected acoustic data during the time of the year when all other sites had low acoustic presence of AMW (March until June).

The overall pattern of high AMW acoustic presence associated with high local sea-ice concentrations is the same for the inner Weddell Sea area. The absence of a latitudinal trend in the inner Weddell Sea area may in part be explained by the distribution of the recorders in the Weddell Sea; at the eastern and western edges of the Weddell Sea, data collection was biased towards lower latitudes, where fewer AMW calls were recorded compared to the higher latitude recording sites, probably due to the differences in ice cover period. Furthermore, differences in ice growth processes in the Weddell Sea may affect how areas with higher sea-ice concentrations (to which AMWs are often associated) are distributed. In the central Weddell Sea, cycles of pancake-ice formation developing into consolidated floes are thought to be the dominant processes by which the sea-ice forms, starting from the centre [39]. By contrast, along the Greenwich meridian, the growing sea-ice cover largely consists of ice platelets, which are formed in the underlying water column in front of the ice-shelf edge, causing sea-ice growth to extend northwards from the continent [39]. The differences in ice growth processes between the GW and WS areas possibly lead to locally denser ice cover in the Weddell Sea, potentially explaining the generally earlier onset of AMW acoustic presence in the WS area compared to most of the GW sites.

The acoustic observations of AMWs overall show a strong association between their acoustic presence and sea-ice concentration during austral winter. Our findings on AMW distribution thereby contrast with the information from visual surveys indicating that AMWs occur primarily within the ice at the sea-ice edge. Visual data collection is, however, by and large limited to the austral summer period and areas with low ice cover. Furthermore, differences in AMW behaviour (e.g. surfacing patterns) when in ice-covered areas may also affect their detectability by visual observers and hence skew distribution information. To further insights on potential seasonal differences in AMW habitat usage, further investigations, e.g. employing animal-borne satellite tags, are clearly needed.

## 4.2. Function of Antarctic minke whale calling

To date little is known on the function of AMW calling behaviour. Across all locations where AMWs have been recorded, acoustic activity is strongly seasonal, albeit with regional differences in the timing of peak calling. Off the Western Antarctic Peninsula, most AMW calls were detected between April and November with peak calling during July [25]. In our study, the peak in AMW acoustic presence throughout the Weddell Sea area occurs from August to October, with few calls occurring throughout the year at the northernmost sites. Recordings of AMWs from lower latitudes on the other hand show that acoustic activity is not restricted to ice-dominated areas and occur during the same period [48,49]. Thomisch et al. [31] reported AMW acoustic presence off Namibia with a double peak: the first peak occurring from June to August and a second peak in November and December. In Perth Canyon (Australia), peak calling also occurred in July–August and again in December [29]. As also concluded by Dominello and Širović [25], these observations clearly falsify previous hypotheses that the bio-duck calls exclusively serve navigational purposes in ice-covered areas [30].

AMW acoustic detections are virtually absent during the summer period when most AMW sightings occur in or close to the marginal sea-ice zone where high density patches of Antarctic krill aggregations are known to occur regularly [50]. Risch et al. [26] recorded bio-duck calls from tagged AMWs that were in large single-species groups of 5 to approximately 40 animals that were feeding almost continuously. Nevertheless, vocalization rates were low; only 32 clear calls of which 6 were bio-duck calls were recorded in this entire dataset. Visual observations from lower latitudes report that AMW breeding behaviour was observed between August and October [51,52]. Catch data furthermore show that most AMW conceptions occur in September [53]. Information collected during winter surveys mentions the presence of AMWs in Antarctic waters but only mature and large juvenile individuals [23,54]. If the assumption that calling behaviour is related to the reproductive activity is correct, our data could suggest that part of AMW breeding may take place in both high-latitude waters with dense ice cover as well as at lower latitude waters. Kasamatsu et al. [55] noted that AMW breeding areas in lower latitude waters may be less concentrated compared to humpback whales (Megaptera novaeangliae) and grey whales (Eschrichtius robustus), which prefer near shore waters for breeding. They hypothesized this may result in differences in meeting probability between mature males and females

compared to species that concentrate in traditional breeding areas in coastal waters. The AMW long calling bouts consisting of repetitive pulses could therefore serve to attract and find mating partners over longer distances. Nevertheless, further work e.g. employing animal-borne tags is needed and underway to further investigate the behavioural contexts in which AMW calls are produced (e.g. [42]).

To date it is unknown how AMW sound production is related to sex, age and reproductive status. Information on staggered and sex-segregated migration in AMWs, presents patchy and partly contrasting information; Kasamatsu *et al.* [55] reported young whales to migrate southbound first, followed by mature and pregnant individuals in October–November. The northbound migration of young animals is thought to start around February, followed by the mature animals in March continuing into early austral winter. Other studies reported female AMWs to prefer higher latitudes during summer [56,57]. Lactating females and calves were rarely observed in Antarctic waters during the summer, whereas pregnant females were found absent from lower latitude waters. Immatures are also thought to avoid high-latitude areas. Laidre *et al.* [58] suggested that off Greenland, in common minke whales (*Balaenoptera acutorostrata*), pregnant females may separate socially to avoid niche overlap or to avoid males by migrating to other areas. Furthermore, occasional sightings of young unweaned common minke whale calves strongly suggest some females calve in more northerly waters outside the breeding areas, possibly skipping migration to low latitudes to calve [59,60].

How these patterns relate to the austral winter situation in Antarctic waters and how calling behaviour relates to sex, age and reproductive status in Antarctic minke whales is not known. However, based on existing evidence, it is possible that the reproductive and conditional status of females determines their migratory strategy.

Although not further quantified here, different bio-duck type calls were observed in our dataset, in accordance to the findings of previous studies that collected similar data in other areas [25,26,29,31,41]. Differences in the acoustic characteristics of bio-duck calls comprised e.g. the duration of the calls, the inter-pulse interval and the number of pulses. Mapping these acoustic differences in the bio-duck calls may shed further light on the function of calling, e.g. whether specific call types are used in different behavioural contexts or if call usage differs in space and time.

## 4.3. Migratory movement

Our data show that, as for many baleen whale populations (i.e. fin (*Balaenoptera physalus*), humpback and Antarctic blue whales (*B. musculus intermedia*); see Geijer *et al.* [61] for a review), AMWs exhibit complex migration patterns. Sightings and passive acoustic recordings show that AMWs are simultaneously present in both low- and high-latitude waters during austral winter [29,31,48]. Off the Western Antarctic Peninsula, visual sightings confirm AMW winter presence with the highest occurrence close to shore and between islands [22,23]. Meanwhile evidence has been accumulating that alternative migration strategies are more the rule than the exception in baleen whales [61]. Both common and Antarctic minke whales remain among the species for which knowledge on migration patterns and wintering habitats is still scarce. Carretta *et al.* [62] indicated the existence of a non-migratory common minke whale population off the west coast of North America, although Risch *et al.* [63] found evidence for a traditional migration pattern in common minke whales. Migration strategies are likely to depend on sex, age and reproductive status as well as ecological factors, and many species are therefore likely to exhibit a repertoire of migratory behaviours [61]. Recent studies using satellite tags on AMWs off the Antarctic Peninsula also suggest movement strategies may differ dramatically between individuals [64]. Tagged AMWs were found to all remain south of the southern boundary of the ACC. One individual travelled along the sea-ice edge and the others remained in the shallower waters of the continental shelf, but all were closely associated with areas with substantial sea-ice coverage [64]. Off South Africa, a bi-modal temporal distribution of AMWs peaking in fall and spring has been observed, suggesting parts of the population migrate to lower latitudes during austral winter [65].

Adaptations to inhabiting heavy sea-ice environments may enable AMWs to exploit this habitat on a year-round basis, potentially limiting the need for migration to low latitude waters to the period when calves are born [66]. The dense sea-ice environment may offer AMWs a refuge from killer whales (*Orcinus orca*), which are known to prey on the species [67]. Their small size enables them to navigate between ice floes, and their strong rostrum allows them to break the ice creating their own breathing holes [22,24]. These adaptations probably enable AMWs to fill a niche and avoid interspecific competition by exploiting krill resources that may not be accessible to other species that also depend on krill as their main food resource [68,69]. Our data, as well, indicate that the traditional migration model of mysticete migration (e.g. [70,71]) is too simplified to describe their life history. However, given that

AMWs seem to produce calls only seasonally, further investigations on their migratory behaviour will need to integrate different methods (i.e. visual and passive acoustic surveys as well as animal-borne tags) to improve our understanding of their migratory repertoire (see [72] for a review).

# 5. Conclusion

The strong relation between AMWs and sea-ice suggests that this species is likely to be sensitive to climate-induced changes to its sea-ice habitat over time. Passive acoustics provide a highly effective tool for monitoring this species during austral winter. Data presented here suggest that the migratory behaviour of this species is more complex than previously thought. Part of the population may undertake seasonal migrations while another part may remain in the ice, with a continuous presence in the Antarctic Southern Ocean during the year. Overall abundance estimates for AMWs are currently around 500 000, whereas earlier assessments estimated 720 000 animals [37] representing a 31% decline. Despite uncertainties and lack of confidence in parts of the assessment, this trend gives reason for concern and recently has led to the classification of AMWs as Near Threatened under the IUCN Red List and under Appendix I of CITES [36]. Japan's decision to leave the IWC could contribute to the recovery of the AMW population in the Southern Ocean. The major difficulties in generating reliable abundance estimates for AMWs are the environment they inhabit [7,20,72]. Clearly, multi-disciplinary approaches are therefore needed to improve our understanding and conserve this elusive species.

Ethics. For this study, we used data deployed and collected from several expeditions, with the *RV Polarstern*. Permission was granted to the Alfred Wegener Institute, Helmholtz-Zentrum für Polar- und Meeresforschung by the Federal Environment Office (Umweltbundesamt UBA). Expedition ANT - XXIV/3 UBA permit no. I 2.4 - 94003-3/207, Expedition ANT - XXV/2 UBA permit no. I 2.4 - 94003-3/217, Expedition ANT - XXVII/2 UBA permit no. I 3.5 -94003-3/255, Expedition ANT - XXVIII/2 UBA permit no. I 3.5 - 94003-3/271, Expedition ANT - XXIX/2 UBA permit no. I 3.5 - 94003-3/286, Expedition ANT- XXX/2 UBA permit no. II 2.8 - 94003-3/324, Expedition PS103 UBA permit no. II 2.8 - 94003-3/38

Data accessibility. Our data are deposited at Dryad: doi:10.5061/dryad.7sqv9s4pd [73].

Authors' contributions. D.F. analysed all the data, conducted statistical analyses and wrote the manuscript. K.T. participated in some data collection and helped draft the manuscript. O.B. participated in collecting data and coordinated the study. T.B. helped guide some statistical analyses and with the previous version of the manuscript. S.S. collected the data. A.Š. helped with the draft manuscript and with guidance for the analyses. I.V.O. coordinated the study, collected part of the data and helped draft the manuscript. All the authors reviewed and contributed to the final document edits. All the authors gave the final approval for publication.

Competing interests. We have no competing interest.

Funding. This research was supported by funding from the Alfred Wegener Institute for Polar and Marine Research. D.F. was funded by the National Agency for Research and Development (ANID) – Chile and the German Academic Exchange Agency (DAAD) grant no. 91643562.

Acknowledgements. Many thanks to the Deutscher Akademischer Austauschdienst (DAAD), Comision Nacional de Investigacion Cientifica y Tecnologia (CONICYT) and the Helmholtz Graduate School for Polar and Marine Research (POLMAR). We also thank the crews of *RV Polarstern* expeditions ANT-XXV/2, ANT-XXIV/3, ANT-XXVII/2, ANT-XXIX/2 and the mooring team of the AWI's physical oceanography department for the deployment and recovery of the acoustic recorders. We also thank Elke Burkhardt and Elena Schall for the many constructive comments and discussions. We also thank all the crew members of the *RV Polarstern*.

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
