## [Reviewer comments · Royal Society Open Science]

Review History

RSOS-192112.R0 (Original submission)

Review form: Reviewer 1 (Denise Risch)

Is the manuscript scientifically sound in its present form?

Yes

Are the interpretations and conclusions justified by the results?

Yes

Is the language acceptable?

Yes

Do you have any ethical concerns with this paper?

No

Have you any concerns about statistical analyses in this paper?

No

Recommendation?

Accept with minor revision (please list in comments)

Comments to the Author(s)

General comments

This manuscript is a well written piece of research drawing together a lot of acoustic recordings to illuminate Antarctic minke whale seasonal distribution and potential migration patterns. The language could be a bit clearer in places and some statements and methods need further explanation.

In general I congratulate the authors to a thorough analysis of a substantial dataset of AMW acoustic presence across a large spatio-temporal scale.

Specific comments

Abstract

I68 This statement indirectly assumes that the bio-duck is related to mating. You are a lot more careful in the rest of the manuscript with this hypothesis, as I think is warranted given the limited knowledge about the behavioural function of the call. I would suggest to rephrase the last sentence of the abstract to make clear that the acoustic presence of AMW in higher latitudes could be due to mating but could also indicate presence of animals for other reasons, i.e. animals simply exploiting foraging opportunities.

Introduction

I107 ...species' behavioural

I109-112 very long sentence – consider splitting it for readability – it is unclear whether you talk about correlation of the species or the species' abundance with sea ice

I116-119 again very long sentence, please revise. ...in particular for....

I124 it is also important to improve this understanding across years

I126 I think it needs another sentence or two to explain why understanding of AMW ecology is important for understanding the whole ecosystem; also be consistent with the wording for spatio-temporal throughout the manuscript you use different version to express the same thing

I129 I was left wanting a bit more on the IWC and CCAMLR mandates for conservation of the Antarctic ecosystem and AMW in particular and also a few sentences on the actual study aims at the end of the discussion

Methods

I143-145 I got a bit confused here. You talk about two sites but compiling data from three recordings?? Also which metrics are you extracting from the other sites? I think this information should come later on where you talk about the acoustic analyses, it seems a bit oddly placed in the data collection section.

I154 did you assess presence at the basis of 1 minute files at a time or over longer timescales, was the resolution always 1 Hz despite changing sample rates? A bit more information on spectrogram parameter settings would be helpful

I155 maybe repeat that the Sonovault data was continuous, to help the reader understand you choice of analyses windows

I162 what is one bio-duck? One phrase? Several phrases?

I174 it would be good to add 1-2 sentences on how these experiments cited were carried out and how the artificial signal matched the bio-duck signal, for example what was the SL of the 'known signal' and is there any SL information for the bio-duck signal?

Results

I191 what do you mean when you say 'acoustic presence was higher'? Do you mean mean relative presence/month?

I193 I assume you mean acoustic presence was low at all sites?

I202 In the central Weddell Sea....

I202 but the missing clear gradient could also be due to the distribution of the recorders in the WS being different from within the GW sector

I221 ...acoustic activity for 11 months...

Discussion

1266 sentence structure – consider revising

1270 but visual surveys during summer are also limited to areas with lower ice cover, so can you really make this statement? Could it not be that visual surveys miss animals in high ice areas, due to longer dive times etc.? I think this is worth a bit more discussion

1293 I agree with this conclusion, but I would like to see more discussion of this hypothesis later on, i.e. just because it is not the only reason for vocalising, could vocalisations not play an important role in navigation under the ice, which would explain the high calling rates during winter?

1296 when you say larger density, what do you mean? Larger compared to what? Please give a bit more background and maybe a citation.

1327 I got a bit confused here – make sure to indicate that this whole paragraph is about the North Atlantic minke whale

1333 maybe compare and contrast with other species too, like humpback whales where PAM and satellite tagging studies are uncovering similar complex patterns of migration

1335 ...migration patterns.

1336-338 revise sentence structure – hard to follow what you want to say here.

1340 ...migratory movement.

1344 a bit more information about what these satellite studies found would be useful

1354 but even if mating happens in winter in ice covered waters – sounds could still be used for other purposes than reproduction, e.g. navigation, which should be discussed a bit further here or earlier on.

1356 unfinished sentence

1357 what differences in repertoire were detected? This should be explained more even if it is still under investigation. As it stands it comes a bit out of the blue and requires more explanation.

1359 replace constant with 'consistent across years' or similar...

1372 there should be more information about current abundance estimates, the recent decision of Japan to abandon the hunt in the Southern Ocean etc. in the Introduction and also a bit more discussion on their current conservation status and the utility of PAM to potentially help with distribution and perhaps, in future, with abundance estimates.

References

1485 and following..adjust order of references

Figures

Caption Figure 1Recording site names...

Caption Figure 2 add the sample rate and resulting time/frequency resolution

Table 2 add information in the caption as to why days were excluded from analysis.

Review form: Reviewer 2

Is the manuscript scientifically sound in its present form?

Yes

Are the interpretations and conclusions justified by the results?

No

Is the language acceptable?

Yes

Do you have any ethical concerns with this paper?

No

Have you any concerns about statistical analyses in this paper?

No

Recommendation?

Accept with minor revision (please list in comments)

Comments to the Author(s)

Please see attached document (Appendix A).

Decision letter (RSOS-192112.R0)

24-Mar-2020

Dear Mr Filun,

The editors assigned to your paper ("Title: Frozen verses: Antarctic minke whales (*Balaenoptera bonaerensis*) call predominantly during austral winter") have now received comments from reviewers. We would like you to revise your paper in accordance with the referee and Associate Editor suggestions which can be found below (not including confidential reports to the Editor). Please note this decision does not guarantee eventual acceptance.

Please submit a copy of your revised paper before 16-Apr-2020. Please note that the revision deadline will expire at 00.00am on this date. If we do not hear from you within this time then it will be assumed that the paper has been withdrawn. In exceptional circumstances, extensions may be possible if agreed with the Editorial Office in advance. We do not allow multiple rounds of revision so we urge you to make every effort to fully address all of the comments at this stage. If deemed necessary by the Editors, your manuscript will be sent back to one or more of the original reviewers for assessment. If the original reviewers are not available, we may invite new reviewers.

- Data accessibility

It is a condition of publication that all supporting data are made available either as supplementary information or preferably in a suitable permanent repository. The data

accessibility section should state where the article's supporting data can be accessed. This section should also include details, where possible of where to access other relevant research materials such as statistical tools, protocols, software etc can be accessed. If the data have been deposited in an external repository this section should list the database, accession number and link to the DOI for all data from the article that have been made publicly available. Data sets that have been deposited in an external repository and have a DOI should also be appropriately cited in the manuscript and included in the reference list.

If you wish to submit your supporting data or code to Dryad (<http://datadryad.org/>), or modify your current submission to dryad, please use the following link:
<http://datadryad.org/submit?journalID=RSOS&manu=RSOS-192112>

- **Competing interests**

- **Authors' contributions**

- **Acknowledgements**

- **Funding statement**

on behalf of the Associate Editor, and Professor Kevin Padian (Subject Editor)
openscience@royalsociety.org

Associate Editor's comments to the Author:

The reviewers are broadly positively inclined towards your paper but each notes substantial

changes that need to be addressed before the paper may be considered for acceptance. Please ensure the changes requested are incorporated into a revised manuscript and also delineated in a point-by-point reply.

Reviewers' Comments to Author:

Reviewer: 1

Comments to the Author(s)

General comments

This manuscript is a well written piece of research drawing together a lot of acoustic recordings to illuminate Antarctic minke whale seasonal distribution and potential migration patterns. The language could be a bit clearer in places and some statements and methods need further explanation.

In general I congratulate the authors to a thorough analysis of a substantial dataset of AMW acoustic presence across a large spatio-temporal scale.

Specific comments

Abstract

168 This statement indirectly assumes that the bio-duck is related to mating. You are a lot more careful in the rest of the manuscript with this hypothesis, as I think is warranted given the limited knowledge about the behavioural function of the call. I would suggest to rephrase the last sentence of the abstract to make clear that the acoustic presence of AMW in higher latitudes could be due to mating but could also indicate presence of animals for other reasons, i.e. animals simply exploiting foraging opportunities.

Introduction

1107 ...species' behavioural

1109-112 very long sentence – consider splitting it for readability – it is unclear whether you talk about correlation of the species or the species' abundance with sea ice

1116-119 again very long sentence, please revise. ...in particular for....

1124 it is also important to improve this understanding across years

1126 I think it needs another sentence or two to explain why understanding of AMW ecology is important for understanding the whole ecosystem; also be consistent with the wording for spatio-temporal throughout the manuscript you use different version to express the same thing

1129 I was left wanting a bit more on the IWC and CCAMLR mandates for conservation of the Antarctic ecosystem and AMW in particular and also a few sentences on the actual study aims at the end of the discussion

Methods

1143-145 I got a bit confused here. You talk about two sites but compiling data from three recordings?? Also which metrics are you extracting from the other sites? I think this information should come later on where you talk about the acoustic analyses, it seems a bit oddly placed in the data collection section.

1154 did you assess presence at the basis of 1 minute files at a time or over longer timescales, was the resolution always 1 Hz despite changing sample rates? A bit more information on spectrogram parameter settings would be helpful

1155 maybe repeat that the Sonovault data was continuous, to help the reader understand your choice of analyses windows

1162 what is one bio-duck? One phrase? Several phrases?

1174 it would be good to add 1-2 sentences on how these experiments cited were carried out and how the artificial signal matched the bio-duck signal, for example what was the SL of the 'known signal' and is there any SL information for the bio-duck signal?

Results

1191 what do you mean when you say 'acoustic presence was higher'? Do you mean mean relative presence/month?

1193 I assume you mean acoustic presence was low at all sites?

1202 In the central Weddell Sea....

1202 but the missing clear gradient could also be due to the distribution of the recorders in the WS being different from within the GW sector

1221 ...acoustic activity for 11 months...

Discussion

1266 sentence structure – consider revising

1270 but visual surveys during summer are also limited to areas with lower ice cover, so can you really make this statement? Could it not be that visual surveys miss animals in high ice areas, due to longer dive times etc.? I think this is worth a bit more discussion

1293 I agree with this conclusion, but I would like to see more discussion of this hypothesis later on, i.e. just because it is not the only reason for vocalising, could vocalisations not play an important role in navigation under the ice, which would explain the high calling rates during winter?

1296 when you say larger density, what do you mean? Larger compared to what? Please give a bit more background and maybe a citation.

1327 I got a bit confused here – make sure to indicate that this whole paragraph is about the North Atlantic minke whale

1333 maybe compare and contrast with other species too, like humpback whales where PAM and satellite tagging studies are uncovering similar complex patterns of migration

1335 ...migration patterns.

1336-338 revise sentence structure – hard to follow what you want to say here.

1340 ...migratory movement.

1344 a bit more information about what these satellite studies found would be useful

1354 but even if mating happens in winter in ice covered waters – sounds could still be used for other purposes than reproduction, e.g. navigation, which should be discussed a bit further here or earlier on.

1356 unfinished sentence

1357 what differences in repertoire were detected? This should be explained more even if it is still under investigation. As it stands it comes a bit out of the blue and requires more explanation.

1359 replace constant with 'consistent across years' or similar...

1372 there should be more information about current abundance estimates, the recent decision of Japan to abandon the hunt in the Southern Ocean etc. in the Introduction and also a bit more discussion on their current conservation status and the utility of PAM to potentially help with distribution and perhaps, in future, with abundance estimates.

References

1485 and following..adjust order of references

Figures

Caption Figure 1 ...Recording site names...

Caption Figure 2 add the sample rate and resulting time/frequency resolution

Table 2 add information in the caption as to why days were excluded from analysis.

Reviewer: 2

Comments to the Author(s)

Please see attached document

Author's Response to Decision Letter for (RSOS-192112.R0)

See Appendix B.

RSOS-192112.R1 (Revision)

Review form: Reviewer 1 (Denise Risch)

Is the manuscript scientifically sound in its present form?

Yes

Are the interpretations and conclusions justified by the results?

Yes

Is the language acceptable?

Yes

Do you have any ethical concerns with this paper?

No

Have you any concerns about statistical analyses in this paper?

No

Recommendation?

Accept as is

Comments to the Author(s)

Well done on the revisions. I have no further comments.

Decision letter (RSOS-192112.R1)

Dear Mr Filun,

It is a pleasure to accept your manuscript entitled "Title: Frozen verses: Antarctic minke whales (*Balaenoptera bonaerensis*) call predominantly during austral winter" in its current form for publication in Royal Society Open Science. The comments of the reviewer who reviewed your manuscript are included at the foot of this letter.

on behalf of Professor Kevin Padian (Subject Editor)
openscience@royalsociety.org

Reviewer comments to Author:

Reviewer: 1
Comments to the Author(s)

Well done on the revisions. I have no further comments.

Appendix A

Review of RSOS-192112

The authors present an impressive data set that should indeed be published and will be helpful in understanding the distribution of this rather enigmatic species. I have just one technical issue that needs to be addressed, i.e., duty-cycling, and is discussed below. However, the discussion about the activity of AMW in different locations in different times of year based just on the occurrence of this call and some sighting records needs to be substantially revised as, for starters, they claim that bio-duck calls have no role in feeding, but the paper that linked AMWs to the bio-duck call occurred with foraging animals, see below. In my humble opinion, the PAM data laying out where these whales occur is incredibly valuable, but to try to assert anything about what the whales are doing during these different times of year in different locations is an unnecessary and unsubstantiated conclusion of the paper. There are ongoing efforts to describe the AMW repertoire and related behavior, which should assist in any further analyses of this impressive data set. But, using just one call, whose function is really unknown, to make larger conclusions about AMW behavior (mating vs. foraging, migration, etc) is not appropriate here.

Line numbers	Comment
139-142	significantly different duty cycles...how does that affect detections and presence calculations? Stanistreet et al (2016) found that duty-cycling can dramatically affect the ability of PAM systems to accurately detect the presence of whales. Their work was on beaked whales and echolocation, which is actually much more predictable than AMW calls, so the duty-cycling issue is perhaps even exacerbated by the lack of information about calling frequency, their own duty cycle, etc. The authors use different time blocks to calculate the LTSAs, but do not indicate whether that was intended to account for the duty-cycling, and, if so, then how?? Furthermore, the authors do have access to some information about cue rates for these calls, as they were reported in the Risch et al. (2014) paper. Cue rates are extremely useful for doing things just like this, estimating the chances of missing a whale based on duty cycles of passive acoustic systems. I was surprised that they had not even mentioned this.
297	The author's statement that 'AMW sounds may not play a role during foraging is at first erroneous and, secondly, the first of several over-reaches in their conclusions based only on the PAM records of one call; they have considered only one type of AMW vocalization, the bio-duck call. It is erroneous as the paper that inextricably linked the bio-duck call with AMW did so with foraging AMWs...I quote from Risch et al (2014)...' The two tags recorded for 18 and 8 h, respectively. During both deployments the tagged whales were in large single-species groups of five to about 40 animals and fed almost continuously [11]. Vocalization

rates were low; only 32 clear calls, with a signal-to-noise ratio of more than 10 dB, were recorded in this entire dataset.' The reference in this short bit of text refers to Friedlaender et al (2014) that reported extraordinarily high feeding rates in these whales. So, the whole argument made by these authors that these calls are associated with social/mating behavior and not foraging must be revised, it is not accurate.

338-340

First of all, Ducklow et al is 2006, not 2007, though presumably this is the paper they intend in the references, but this paper says nothing about sighting locations nor seasonality of AMW sightings. Furthermore, neither of these papers reports acoustic data, so AMW calling behavior is not covered. Lastly, I don't believe either study reported any survey effort between Sept and Feb, which leaves a significant gap in this argument.

355-361

this mixture of quasi-results and speculation have no business being in this paper. 'acoustic groups' or 'subpopulations' are simply speculations, especially since the authors just reaffirmed that movement patterns appear to be variable and individually specific (lines 342-344) and that they have investigated only one type of call and presented no data about the variability/characteristics of that one call nor data on any other calls. AMW vocal repertoire certainly needs to be investigated, but a few words here is not appropriate. Indeed the repertoire is being investigated and described from tag data collected on nearly 30 AMW, and while it's only a conference abstract so far, the authors should be aware of this paper, i.e., Weindorf et al. (2019).

Friedlaender, A. S., Goldbogen, J. A., Nowacek, D. P., Read, A. J., Johnston, D., & Gales, N. (2014). Feeding rates and under-ice foraging strategies of the smallest lunge filter feeder, the Antarctic minke whale (*Balaenoptera bonaerensis*). *Journal of Experimental Biology*, 217(16), 2851-2854.

Risch, D., Gales, N. J., Gedamke, J., Kindermann, L., Nowacek, D. P., Read, A. J., ... & Friedlaender, A. S. (2014). Mysterious bio-duck sound attributed to the Antarctic minke whale (*Balaenoptera bonaerensis*). *Biology letters*, 10(4), 20140175.

Stanistreet, J. E., Nowacek, D. P., Read, A. J., Baumann-Pickering, S., Moors-Murphy, H. B., & Van Parijs, S. M. (2016). Effects of duty-cycled passive acoustic recordings on detecting the presence of beaked whales in the northwest Atlantic. *The Journal of the Acoustical Society of America*, 140(1), EL31-EL37.

Weindorf, S. et al. 2019. Behavioral and environmental context of Antarctic minke whale vocalizations. World Marine Mammal Conference, Barcelona, December 2019.

Appendix B

Review of RSOS-192112

Reviewer 1:

General comments

This manuscript is a well written piece of research drawing together a lot of acoustic recordings to illuminate Antarctic minke whale seasonal distribution and potential migration patterns. The language could be a bit clearer in places and some statements and methods need further explanation.

In general I congratulate the authors to a thorough analysis of a substantial dataset of AMW acoustic presence across a large spatio-temporal scale.

Specific comments

Abstract

168 This statement indirectly assumes that the bio-duck is related to mating. You are a lot more careful in the rest of the manuscript with this hypothesis, as I think is warranted given the limited knowledge about the behavioural function of the call. I would suggest to rephrase the last sentence of the abstract to make clear that the acoustic presence of AMW in higher latitudes could be due to mating but could also indicate presence of animals for other reasons, i.e. animals simply exploiting foraging opportunities.

- We agree and rephrased this sentence to: ‘The period with highest acoustic presence in the Weddell Sea (Sep-Oct) coincides with the timing of the breeding season of AMW in lower latitudes. The bio-duck call could therefore play a role in mating, although other behavioral functions cannot be excluded to date.’ **Lines:66-68**

Introduction

1110 ...species’ behavioural

- Changed into ‘species’ behavior and appearance’.

1109-112 very long sentence – consider splitting it for readability – it is unclear whether you talk about correlation of the species or the species’ abundance with sea ice

- According to the reviewer’s suggestions we split this sentence in two sentences:
Lines:109-112 ‘As a consequence, AMW abundance estimates based on visual survey data are in many cases spatially and temporally biased (Thiele *et al.* 2004; Williams *et al.* 2014). Only very sparse information exists on the winter distribution of AMW in the Southern Ocean and how the species is associated to different sea ice concentrations throughout the year (Aguayo-Lobo 1994; Ainley *et al.* 2012a; Dominello and Širović 2016).’

1116-119 again very long sentence, please revise. ...in particular for....

- According to the reviewer’s suggestions we split this in two sentences:
‘This makes passive acoustic recording techniques a suitable tool to remotely monitor acoustic presence and study marine mammal behavior. This particularly accounts for polar species given that large parts of their habitats are (seasonally) inaccessible for research vessels (e.g., Širović *et al.* 2004; Van Opzeeland *et al.* 2013; Thomisch *et al.* 2016; Stafford *et al.* 2018).’ **Lines:116-119**

1128 it is also important to improve this understanding across years

- Changed accordingly. **Lines:124**

1126 I think it needs another sentence or two to explain why understanding of AMW ecology is important for understanding the whole ecosystem; also, be consistent with the wording for spatio-temporal throughout the manuscript you use different version to express the same thing

- According to the reviewer's suggestions we extended this motivation. We also changed 'spatio-temporal' into 'spatial-temporal' throughout the manuscript. **Lines:125-139**

1129 I was left wanting a bit more on the IWC and CCAMLR mandates for conservation of the Antarctic ecosystem and AMW in particular and also a few sentences on the actual study aims at the end of the discussion.

- According to the reviewer's suggestions we added a few sentences about the conservation mandates from the CCAMLR and IWC here and to the Conclusion. **Lines:136-142**
- **Discussion lines417-423**

Methods

1167-173 I got a bit confused here. You talk about two sites but compiling data from three recordings?? Also, which metrics are you extracting from the other sites? I think this information should come later on where you talk about the acoustic analyses, it seems a bit oddly placed in the data collection section.

- We have two positions where we record with three recorders deployed in the same mooring. Unfortunately, the 3 recorders didn't collect data during the whole year. We therefore decided to compile the data from the three recorders from the same mooring to obtain one effective year of the mooring position. We have tried to clarify this in the manuscript and inserted this information in the results section as suggested by the reviewer (**Lines 119-122**). We extracted the same metrics from all sites.

1154 did you assess presence at the basis of 1-minute files at a time or over longer timescales, was the resolution always 1 Hz despite changing sample rates? A bit more information on spectrogram parameter settings would be helpful.

- The resolution of the data for analysis was kept at 1 Hz. Daily presence was assessed for one-day of data in case of the continuous SonoVault recordings and 4-day windows for the duty cycled recordings 5 minutes in case of the subsampled AURAL recordings. We have tried to clarify this better in the text and also added information on the spectrogram parameters settings that were used. **Lines:169-175**

1155 maybe repeat that the Sonovault data was continuous, to help the reader understand your choice of analyses windows.

- **Lines171:** Changed accordingly.

1162 what is one bio-duck? One phrase? Several phrases?

- We used the definition of Risch et al. 2014 and Dominello & Širović 2016 to identify a bio-duck call. The bio-duck is characterized by its repetitive nature, consisting of regular down-sweeps or pulses, with most energy located in the 50-300 Hz band. The number of pulses can vary but occurs in an interval <1 s (Dominello & Širović 2016). In our study bio-duck calls never occurred alone (i.e. one cluster of calls, or one phrase). The minimum sequence duration that we observed was 20s. Days with AMW acoustic presence therefore at least had a series of 20 seconds with bio-duck calls. **Lines:176-186**

1174 it would be good to add 1-2 sentences on how these experiments cited were carried out and how the artificial signal matched the bio-duck signal, for example what was the SL of the 'known signal' and is there any SL information for the bio-duck signal?

- As the reviewer suggested, we now included more information about the SLs used for calculating the AMWs sound propagation. The detection range was estimated based on the sound propagation of a known signal between 259 and 261 Hz with source levels varying between 171- 174 [dB re 1µPa] emitted by oceanographic instruments deployed in the Weddell Sea. The calculated transmission loss from this empirical analysis was used to estimate the maximum distance, an AMW downsweep call (with SL = 147 dB re 1µPa; *Risch et al.* 2014a) would travel and would still be visually detectable in a spectrogram.

To account for any uncertainties the higher SL from Risch et al. (2014a) was used and AMW vocalizations were assumed to be detectable within a radius of 40 km, rather than the calculated 27 km. Previous studies on common minke whale vocalizations describe SLs in the range of 160-165 dB re 1 µPa (Winn and Perkins 1976; Gedamke 2001). For AMW bio-duck calls, RLs = 140.2 +/- 3.6 dB re 1 µPa were reported (recorded with animal-borne tags) with most energy located between 50-300 Hz, but only 6 samples were recorded in this study and all calls stemmed from one individual (Risch et al. 2014a). **Lines:195-205**

Results

1191 what do you mean when you say '?' Do you mean mean relative presence/month?

- We clarified this throughout the text. Yes, with this we refer to 'relative presence/month'. **Lines:227**

1193 I assume you mean acoustic presence was low at all sites?

- Yes, this is correct. Changed accordingly.**line:230**

1202 In the central Weddell Sea...

- Changed accordingly**Lline:239**

1202 but the missing clear gradient could also be due to the distribution of the recorders in the WS being different from within the GW sector

- Yes, this is correct. We included a sentence in the discussion to clarify this point. **Lines:305-310**

1221 ...acoustic activity for 11 months...

- Changed accordingly.**Line:258**

Discussion

1266 sentence structure – consider revising

- Sentence structure revised. **Lines:304-305**

1270 but visual surveys during summer are also limited to areas with lower ice cover, so can you really make this statement? Could it not be that visual surveys miss animals in high ice areas, due to longer dive times etc.? I think this is worth a bit more discussion

- We fully agree with the reviewer's suggestion and have modified this section accordingly. **Lines:319-324**

1293 I agree with this conclusion, but I would like to see more discussion of this hypothesis later on, i.e. just because it is not the only reason for vocalising, could vocalisations not play an important role in navigation under the ice, which would explain the high calling rates during winter?

- Based on the comments of the second reviewer that some of our hypotheses were too far-stretched given on the actual data we have removed the section where we discuss potential behavioral function of the bio-duck call. We would like to point out that we already have a long section in the discussion where we describe the potential function of calls during feeding, navigation and mating behaviors. Here, we also compare our results with previous studies who describe acoustic patterns of AMWs in other (ice-free) regions i.e., Australia, and Namibia. **Lines:226-277**

1296 when you say larger density, what do you mean? Larger compared to what? Please give a bit more background and maybe a citation.

- This sentence was changed according to the reviewer's question into: 'AMW acoustic detections are virtually absent during the summer period when most AWM sightings occur in or close to the marginal sea-ice zone where high density patches of Antarctic krill aggregations are known to occur regularly (Lascara *et al.* 1999)' **Lines:339-341**

1327 I got a bit confused here – make sure to indicate that this whole paragraph is about the North Atlantic minke whale.

- Changed accordingly. **Lines:360-366**

1333 maybe compare and contrast with other species too, like humpback whales where PAM and satellite tagging studies are uncovering similar complex patterns of migration

- As the reviewer suggest we include more examples about other whales species with similar complex migratory patterns i.e., humpback and Antarctic blue whales (Van Opzeeland *et al.*, 2013, Thomisch *et al.* 2016) as well as reference to a highly relevant review paper by Geijer *et al.* 2016. **Lines:379-371**

1335 ... migration patterns.

- Changed accordingly. **Line:381**

1336-338 revise sentence structure – hard to follow what you want to say here.

- Sentence revised. **Line 381-382**

1340 ...migratory movement.

- Sentence removed.

1344 a bit more information about what these satellite studies found would be useful.

- We included more information from the Lee *et al.* 2017 study. **Lines: 393-395**

1354 but even if mating happens in winter in ice covered waters – sounds could still be used for other purposes than reproduction, e.g. navigation, which should be discussed a bit further here or earlier on.

- See previous reply to Reviewer 1 comment to **Line: 239**.

1356 unfinished sentence

- Changed accordingly **Line: 405-410**

1357 what differences in repertoire were detected? This should be explained more even if it is still under investigation. As it stands it comes a bit out of the blue and requires more explanation.

- The requested information has been included, but the section has been shortened to also accommodate the concerns of reviewer 2. **Lines:372-377**

1359 replace constant with 'consistent across years' or similar....

- Sentence rephrased. **Lines:374-377**

1372 there should be more information about current abundance estimates, the recent decision of Japan to abandon the hunt in the Southern Ocean etc. in the Introduction and also a bit more discussion on their current conservation status and the utility of PAM to potentially help with distribution and perhaps, in future, with abundance estimates.

- Additional information included in the Conclusion, see **Lines:417-426**

References

1485 and following..adjust order of references.

- References were checked and adjusted

Figures

Caption Figure 1Recording site names....

- Recording site names are in the caption.

Caption Figure 2 add the sample rate and resulting time/frequency resolution

- Information added to the caption of Fig 2.

Table 2 add information in the caption as to why days were excluded from analysis.

- Information added to the caption of Table 2.

Reviewer 2:

The authors present an impressive data set that should indeed be published and will be helpful in understanding the distribution of this rather enigmatic species. I have just one technical issue that needs to be addressed, i.e., duty-cycling, and is discussed below.

However, the discussion about the activity of AMW in different locations in different times of year based just on the occurrence of this call and some sighting records needs to be substantially revised as, for starters, they claim that bio-duck calls have no role in feeding, but the paper that linked AMWs to the bio-duck call occurred with foraging animals, see below. In my humble opinion, the PAM data laying out where these whales occur is incredibly valuable, but to try to assert anything about what the whales are doing during these different times of year in different locations is an unnecessary and unsubstantiated conclusion of the paper. There are ongoing efforts to describe the AMW repertoire and related behavior, which should assist in any further analyses of this impressive data set. But, using just one call, whose function is really unknown, to make larger conclusions about AMW behavior (mating vs. foraging, migration, etc) is not appropriate here.

➤ **Reply to general comment Reviewer 2:**

We thank the reviewer for these constructive comments and compliments. As to the duty-cycling issue, the revised version of the manuscript now includes an additional analysis of a subset of the data to substantiate our decision on treating the subsampled and continuous recordings similarly with respect to the interpretation of AMW daily acoustic presence. See also detailed reply further below.

Furthermore, to overcome a potential misunderstanding, we would like to clarify that we did not use one bio-duck *call type* in our analyses, but included all bio-duck types to assess overall AMW daily acoustic presence. As also briefly addressed in the manuscript, we recognized differences in the bio-duck call in our data compared to previous work by Dominello & Sirovic (2016) and we are analyzing this in further detail to assess spatio-temporal patterns in AMW repertoire composition. However, given that this is a work in progress and also is beyond the scope of the current manuscript, we decided for this analysis to solely assess acoustic presence based on the presence of *any* bio-duck call in the data. We have also clarified this further in the text. **Lines 176-186**

We understand the point that the reviewer makes regarding our hypotheses on the behavioral function of the bio-duck calls and have removed the - admittedly quite strong -conclusions in various parts of the manuscript on the potential reproductive function of AMW calls from the discussion. Instead, we included the suggested information on the feeding context during which the calls from Risch et al. (2014a) were recorded. Nevertheless, it needs to be kept in mind that Risch et al. (2014a) also reported that calling rates were low (total 38 calls over a 26-hour recording period of two tags, during which only 6 of the 38 calls were bio-duck calls) despite the fact that animals were in groups of up to 40 animals. The presence of bio-duck calls during feeding should therefore also not be over-interpreted. Occasional singing behavior on the feeding grounds seems to be a common feature of many humpback whale populations worldwide. If for AMWs, the bio-duck calls serve a function during reproduction, it is conceivable that they may also sparsely be recorded during feeding.

In the text, we maintained the brief summary of available contextual information on AMW migratory movements, while clearly recognizing that there are considerable gaps in the data that need to be filled before further conclusions on movement patterns can be drawn. This section also serves to clarify that our data add to the growing amount of evidence that migratory behavior of baleen whales for virtually all species is far more complex than previously assumed.

Line numbers Comment

139-142 significantly different duty cycles...how does that affect detections and presence calculations?
Stanistreet et al (2016) found that duty-cycling can dramatically affect the ability of PAM systems to accurately

detect the presence of whales. Their work was on beaked whales and echolocation, which is actually much more predictable than AMW calls, so the duty-cycling issue is perhaps even exacerbated by the lack of information about calling frequency, their own duty cycle, etc. The authors use different time blocks to calculate the LTSAs, but do not indicate whether that was intended to account for the duty-cycling, and, if so, then how??

Furthermore, the authors do have access to some information about cue rates for these calls, as they were reported in the Risch et al. (2014) paper. Cue rates are extremely useful for doing things just like this, estimating the chances of missing a whale based on duty cycles of passive acoustic systems. I was surprised that they had not even mentioned this.

- We agree with the reviewer that this issue should have been clearly addressed in the manuscript to exclude introducing any biases by differences in sampling regimes. To solve this in the revised version, we included an additional sub-analysis of one year of continuous data (see Appendix text for recorder details), which was analysed on an hourly basis for AMW acoustic presence. To identify the variability to use different duty cycles, we created a sub-sampled dataset mimicking the two different duty-cycles of the AURAL devices, i.e., 5 min every 4 hours and 5 min every hour. In the Appendix, we describe that the general acoustic pattern is slightly underestimated by both duty cycles (number of days with acoustic presence per month), but that the overall acoustic presence pattern on a monthly basis is maintained. The differences in duty-cycles is obviously largest during the shoulder season, when calling activity is just increasing or decreasing. Furthermore, we are not aware that any reliable and truly representative cue rate information can be extracted from the Risch et al. (2014a) paper, given that only 6 bio-duck calls were recorded there that stem from one individual.

297 The author's statement that 'AMW sounds may not play a role during foraging is at first erroneous and, secondly, the first of several overreaches in their conclusions based only on the PAM records of one call; they have considered only one type of AMW vocalization, the bio-duck call. It is erroneous as the paper that inextricably linked the bio-duck call with AMW did so with foraging AMWs...I quote from Risch et al (2014)...' The two tags recorded for 18 and 8 h, respectively. During both deployments the tagged whales were in large single-species groups of five to about 40 animals and fed almost continuously [11]. Vocalization rates were low; only 32 clear calls, with a signal-to-noise ratio of more than 10 dB, were recorded in this entire dataset.' The reference in this short bit of text refers to Friedlaender et al (2014) that reported extraordinarily high feeding rates in these whales. So, the whole argument made by these authors that these calls are associated with social/mating behavior and not foraging must be revised, it is not accurate.

- This comment is in line with some of the comments of reviewer 1. See therefore also replies to reviewer 1. We agree and removed as well as rephrased parts throughout the manuscript to modify according to this comment. Lines:326-337

338-340 First of all, Ducklow et al is 2006, not 2007, though presumably this is the paper they intend in the references, but this paper says nothing about sighting locations nor seasonality of AMW sightings. Furthermore, neither of these papers reports acoustic data, so AMW calling behavior is not covered. Lastly, I don't believe either study reported any survey effort between Sept and Feb, which leaves a significant gap in this argument.

- In response to this, we replaced the reference for a new one that indeed better fits the context of our discussion. In the text we describe that AMWs as other whale species, seem to exhibit a complex migration pattern, in which part of the population possibly remains in Antarctic waters year-round. To include further observations from other studies, we added information about the presence of AMWs during autumn and winter based on visual and acoustic information. We included Thiele et al. 2004; and the report of Aguayo Lobo 1994 as references for AMW sightings during austral winter surveys. In addition, we included the study of Dominello and Širović 2016, describing the year-round acoustic behavior of AMWs off the WAP, reporting their acoustic presence from May to November with peak calling in July. **Lines:383-384**

355-361 this mixture of quasi-results and speculation have no business being in this paper. 'acoustic groups' or 'subpopulations' are simply speculations, especially since the authors just reaffirmed that movement patterns appear to be variable and individually specific (lines 342-344) and that they have investigated only one type of call and presented no data about the variability/characteristics of that one call nor data on any other calls. AMW vocal repertoire certainly needs to be investigated, but a few words here is not appropriate. Indeed the repertoire is being investigated and described from tag data collected on nearly 30 AMW, and while it's only a conference abstract so far, the authors should be aware of this paper, i.e., Weindorf et al. (2019).

- We have shortened this section and removed speculations on acoustic groups and subpopulations. Furthermore, we have included the recent study by Shabangu et al. (2020) as well as the work by Weindorf et al. (2020) presented at the last WM conference, which indeed will be very helpful to improve our understanding of AMW calling behavior.